# Pediatric Massage Therapy Research: A Narrative Review

**DOI:** 10.3390/children6060078

**Published:** 2019-06-06

**Authors:** Tiffany Field

**Affiliations:** University of Miami/Miller School of Medicine, Fielding Graduate University, 2889 McFarlane Rd, Miami, FL 33133, USA; tfield@med.miami.edu

**Keywords:** pediatric massage therapy, infant massage, child massage, adolescent massage

## Abstract

This narrative review on pediatric massage literature from the last decade suggests that massage therapy has positive effects on several pediatric conditions. These include preterm infant growth, psychological problems including aggression, gastrointestinal problems including constipation and diarrhea, painful conditions including burns and sickle cell, muscle tone disorders including cerebral palsy and Down syndrome, and chronic illnesses including diabetes, asthma cancer, and HIV. Potential underlying mechanisms for the massage therapy effects include increased vagal activity and decreased stress hormones. Limitations of the literature include the need for more randomized controlled trials, longitudinal studies, and underlying mechanism studies.

## 1. Introduction

The pediatric massage therapy research literature was very prolific for many years, but it seems to have been diminishing over the last several years. This narrative review is derived from literature searches of PubMed and PsychINFO for publications from the last decade (2008–2018) that have the terms pediatric massage or massage and children in their titles. The review includes randomized controlled trial studies, systematic reviews and meta-analyses. Pilot studies and foreign language papers were excluded. Of the 139 papers reviewed, only 48 met the inclusion criteria to be summarized here. Studies from the previous two decades are cited as supportive material.

Massage therapy has shown positive effects for pregnancy and labor, for preterm infant growth, for psychological problems including aggression, anxiety, depression, and posttraumatic stress disorder, for gastrointestinal problems including constipation and diarrhea, for painful procedures and pain syndromes including burns and sickle cell, for muscle tone/movement disorders including cerebral palsy and Down syndrome, and for chronic illnesses including diabetes, asthma, cancer, and HIV. Potential underlying mechanisms for the massage therapy effects are discussed including increased vagal activity. Limitations of the literature are also reviewed including the need for more randomized controlled trials and longitudinal studies. The paper is accordingly divided into sections on each of these topics. The effects reported for the studies are typically based on assessments made at the end of the studies unless otherwise noted.

## 2. Preterm Newborn Massage Therapy

Preterm infant massage therapy effects have been researched for several decades and have been reviewed by many including our reviews of 2010 and 2018 [1,2]. Our earlier review was focused primarily on massage therapy for the reduction of procedure-related pain in preterm newborns, for example, pain associated with heelsticks [3]. Other earlier newborn massage studies reported better developmental scores for massaged versus control preterm newborns who had been exposed to cocaine [4] or to HIV in utero [5]. In the latest review covering the last decade, fifteen papers were cited for preterm newborn massage therapy leading to greater weight gain, three for less infection, six for shorter hospital stays, six for developmental gains and three for reduction of parental stress [6]. In another recent review of randomized controlled trials, the studies revealed not only greater weight gain and earlier discharge but also greater pain tolerance [7]. Further, as reported in that review, better developmental scores and immune function were also noted in several longitudinal studies on preterm infants who had been given the Field et al. [8] moderate pressure massage protocol [8]. Similarly, in a meta-analysis on 34 preterm infant massage studies conducted in many countries, the massaged preterm newborns averaged significantly greater weight gain (20 versus 16 g per day) and a shorter hospital stay (27 versus 31 days) [9]. Once again, the moderate pressure Field et al. [8], (1986) protocol was used in these studies.

Some of these effects were confounded by the oils used for the massage therapy. This was especially notable in the study that used medium-chain triglyceride oil which resulted in a significantly greater weight gain when that oil was used as compared to a group who received massage alone and a non-massage control group [10]. The authors referred to this effect as “transcutaneous feeding”. This has been noted for other oils as well including a comparison between sunflower oil massage versus massage alone [11]. In addition, some oils have been more effective than others as, for example, in a study that showed coconut oil to be more effective than sunflower oil for preterm newborns [12]. Nonetheless, despite the added effect of oils in all of these comparisons, the massage alone group has fared better than the non-massage group on greater weight gain and shorter hospital stay.

In addition to the significant clinical gains shown following massage therapy of preterm infants, several studies have documented the safety of preterm massage as well as potential underlying mechanisms that may inform its positive effects. For example, even though temperature might be expected to decrease while the incubator portholes remained open for the massage protocol, temperature increased for those preterm infants receiving massage [13]. Potential mechanism studies have included increased vagal activity and gastric motility following massage, leading to increased weight gain [14]. This effect was later documented as unique to massage therapy [15]. In this study, massaging preterm infants versus exercising their limbs led to increased weight gain via increased vagal activity. The exercise-associated weight gain was related to increased calorie consumption. Still another underlying mechanism was suggested by increased serum levels of insulin and insulin-like growth factor-1 even after only five days of moderate pressure massage [16]. Moreover, most importantly, another research group reported increased natural killer cell activity following massage, which would lead to fewer bacterial and viral infections in preterm newborns [17]. Surprisingly, despite these important physiological/biochemical effects, other research groups have continued to assess the weight gain and hospital stay measures as opposed to potential underlying mechanism measures, probably because of limited research funds. Further, more studies using mothers as therapists would help document the cost-effectiveness of preterm infant massage and promote the wider use of massage in neonatal intensive care units [18].

## 3. Full-Term Newborn Massage Therapy

Fewer massage therapy studies have been conducted on full-term infants, likely because full-term infants are hospitalized for shorter periods of time and because they have less need of massage therapy, for example, for weight gain. Most of the full-term newborn massage studies to date have been focused on decreasing hyperbilirubinemia and reducing irritability and sleep disturbances which are the problems most frequently reported to pediatricians by parents. In a recent review of this literature, five studies showed that massage reduced bilirubin levels, three that it decreased sleep disturbances, four that it reduced colic/crying, three that it improved interactions, one that it facilitated development, and three that it reduced parent stress [19]. 

Since that review was published, only a meta-analysis could be found on massage reducing bilirubin levels in full-term newborns [20]. This analysis included six randomized controlled trials involving 357 infants. The analysis suggested that transcutaneous bilirubin levels were significantly reduced after four days of massage when massage groups were compared to control groups. These results were not surprising inasmuch as the studies on hyperbilirubinemia in the previous review were included in this meta-analysis and they had also suggested that by the fourth day bilirubin levels were reduced. Once again, these studies involved moderate pressure massage, i.e., moving the skin. A potential underlying mechanism for these effects is increased vagal activity leading to enhanced gastric motility that, in turn, would reduce bilirubin levels, although these measures were not recorded in these studies. Future studies might also use mothers as therapists given that at least one of these studies showed positive effects of mothers providing the therapy.

Mothers have been taught to massage their infants. For example, in one study, massage by mothers enhanced the circadian sleep rhythms of their infants [21]. In another study, mothers were asked to massage their newborns for 10 min prior to bedtime on a daily basis for one month [22]. The infants’ sleep became more organized as suggested by more prolonged periods of sleep, fewer night wakings and more hours of sleep. Not surprisingly, the mothers’ sleep also became more organized, probably because their infants were sleeping instead of awakening them and possibly because they were also experiencing the benefits of pressure stimulation in their hands from giving the massages. The positive effects of massaging on the massager have been noted in earlier studies on slightly older infants, e.g., elderly adults massaging infants [23], teachers massaging infants [24] and fathers massaging their infants [25], suggesting that the stimulation of their hands may be a key factor. Further research is needed on potential underlying mechanisms for the effects of massage on the massager as well as the massage.

## 4. Child and Adolescent Massage Therapy

Dozens of studies have been conducted on massage therapy effects on children and adolescents over the last few decades. Because these have not been reviewed in one place previously, they will be referenced here. The child and adolescent massage therapy studies have focused almost exclusively on problematic conditions including attention disorders, psychological problems, gastrointestinal problems, motor tone disorders, pain syndromes, autoimmune and immune conditions. Accordingly, these studies are grouped by these categories in the following summaries.

### 4.1. Attention Disorders

Attention disorders have responded positively to massage therapy including autism spectrum disorder (ASD) and attention deficit hyperactivity disorder (ADHD). In one of the first studies on ASD, children who were given 15-min daily massages by therapists for a month showed more attentive behavior in the classroom [26]. This was subsequently followed by a study on similar young children with ASD who were massaged by their parents for 15-min daily before bedtime [27]. After one month of daily massages, the children were not only more attentive at school but also had fewer sleep problems, which may have contributed to their greater attentiveness. 

Children and adolescents with ADHD have also been helped by massage therapy. After receiving two twenty-minute massages per week for four weeks, their attentiveness was increased and their hyperactivity decreased based on teacher observations in at least two studies [28]. 

Given the pervasiveness of these disorders and their serious effects on social and cognitive development, and inasmuch as massage therapy has significantly increased attentiveness, it is not clear why so few massage therapy studies have been conducted on children with attention problems. The reduction in sleep problems for the ASD children suggests a potential underlying mechanism for their increased attentiveness. Although vagal activity and EEG were not measured in these studies, they are also potential underlying mechanisms. Vagal activity increases following massage therapy, and it is associated with increased attentiveness [29] as well as increased alpha and beta activity and more accurate math computations [30]. 

### 4.2. Psychological Problems

Psychological problems have also been the focus of pediatric massage therapy research including aggression, anxiety, depression, and posttraumatic stress disorder (PTSD). In a study on preschoolers “with aggressive and deviant behavior” 50 children were massaged daily for 20-min and compared to a group of 50 children who listened to stories [31]. By the third month and again at 6 and 12 months, the massaged children had lower parent and teacher rated aggression scores on the Child Behavior Checklist. The children also had fewer social and somatic problems. 

In a similar paradigm used by another group of researchers, three groups of second grade children including massage, storytelling and control groups were compared on their aggressive behavior and academic performance after ten fifty-minute sessions [32]. In this case, both the massage and storytelling groups showed decreased aggressive behavior and improved academic performance, suggesting that giving children extra attention in itself helps reduce their aggression. 

In an earlier study, aggressive adolescents who were residing on a closed hospital unit because of their aggressive behavior were given a month of 10-min daily chair massages [33]. At the end of the month they were less aggressive. In a similar study on children and adolescents who were hospitalized for their anxiety and depression, those mood disorders were not only decreased but other measures of anxiety and depression including norepinephrine and cortisol were decreased following a month of 20-min daily massages [34]. Hospitalization anxiety itself has been reduced in school-age children who received 20-min massages three times a day for three days [35]. On the fifth day, their state anxiety as well as their heart rate and blood pressure were reduced. 

Following Hurricane Andrew, children who had experienced the eye of the hurricane showed posttraumatic stress disorder symptoms [36]. Following one month of daily massages, they had fewer PTSD symptoms including less depression, anxiety, and sleep disturbance. In addition, their self-drawings (a very reliable measure for children) were more colorful and more elaborate than at the beginning of the study.

Eating disorders including anorexia and bulimia might also be considered psychological disorders for their associated depression symptoms. Eating disorders have rarely been treated by massage therapy despite the notable decrease in depression following massage. In an early study on anorexia in adolescent girls, eating disorder symptoms and depression were decreased and excessively low-dopamine levels were increased following one month of daily 20-min massages [37]. The same effects were noted for adolescent girls with bulimia following the same protocol in a separate study [38]. These studies highlight the importance of assessing massage therapy effects on biochemical imbalances such as depressed dopamine levels as well as behavior and mood disturbances in children and adolescents with psychological problems.

### 4.3. Gastrointestinal Problems

Gastrointestinal problems including both constipation and diarrhea have been reduced by massage therapy. In a study on children with chronic constipation, parents were asked to provide abdominal massage for 20 min per day for a month [39]. Several improvements occurred including reduced constipation symptoms (88%) and laxative medication use (58%) as well as improved diet (41%). Diarrhea has also been traditionally treated by massage given by Chinese medicine physicians. In a meta-analysis of 26 studies including 2644 children with acute diarrhea, pediatric massage was significantly and clinically more effective than pharmacotherapy [40]. However, databases that were based on massage alone were combined with those that were derived from multiple complementary therapies in this meta-analysis. 

While constipation and diarrhea are clearly opposing conditions, their improvement following massage therapy may paradoxically relate to a common underlying mechanism including increased vagal activity (via the vagal branch leading to the gastrointestinal tract) resulting in increased gastric motility. Although not measured in these studies, both vagal activity and gastric motility have increased following massage in infants [14].

### 4.4. Motor Tone Disorders

Two other contrasting disorders are cerebral palsy for its excessive motor tone (spasticity or hypertonicity) versus Down syndrome for its hypotonicity (flaccidity). Massage therapy has reduced both hypertonicity and hypotonicity.

In early studies on cerebral palsy, children’s hypertonicity was decreased after a month of daily bedtime massages by their parents [41,42]. In a more recent study from Islamabad, Pakistan, children with cerebral palsy were given 30-min massages five times per week for six weeks [43]. The children’s spasticity was significantly reduced by the end of the study. In a survey study on the prevalence of massage therapy for children and adolescents with cerebral palsy, 80% of the children had received massage while only 51% were currently receiving massage and 77% by their parents [44]. 

Down syndrome infants with hypotonicity showed increased muscle tone following a month of 15-min daily massages by their mothers [45]. In that study, motor scores and object manipulation scores improved in both the children with hypertonicity and hypotonicity following a 5-month massage therapy period [45]. Improvement continued at a 6-month follow-up assessment even though massage therapy effects do not typically persist when the therapy is discontinued.

Children with motor development delays but with no cerebral palsy diagnosis have also benefited from massage therapy [46]. In this study, children 1–3 years of age were randomly assigned to a control group or a massage group that received 20 min massages twice a week for 12 weeks. Unlike most other studies on children with motor development problems, these researchers stratified the children by age and motor development quotient, making this massage study more robust than most. The massage group showed greater improvement on gross motor and sensory sensitivity scores.

### 4.5. Painful Procedures and Pain Syndromes

Pediatric massage has reduced pain during painful procedures as well as for chronic pain conditions. Painful procedures have included hematopoietic cell transplantation and cardiovascular surgery. Burns and post burn pain have been the focus of several pediatric massage studies. Chronic pain in general and juvenile rheumatoid arthritis pain in particular have also been reduced by massage therapy.

In a study on hematopoietic cell transplantation, children received massages from both therapists and parents [47]. The children reputedly experienced pain relief from the massage. In addition, they had less nausea and they had an easier time falling asleep. The parents also reported increased closeness to their children following massage. The parents preferred a semi-standardized massage protocol, highlighting the individual preferences for different massage techniques. Although the parents were requested to apply moderate pressure, there were no compliance checks on the parents’ massages in this study. In a study on pain and anxiety after cardiovascular surgery, sixty children were randomly assigned to receive three massage sessions or three reading visits on the first three postoperative days [48]. The children receiving massage therapy had significantly lower anxiety scores and lower total benzodiazepine exposure for the three days following heart surgery.

In one of the first massage studies on young children with burns, the children’s pain thresholds were increased following 20-min massages as suggested by less distress and crying during their subsequent skin-brushing sessions [49]. In a later study, children from the same burn unit were noted to have less post-burn pain and itching following 20-min massages [50]. As would be expected, decreased crying in children with burns has been accompanied by a significant decrease in heart rate and respiration following massage [51]. In addition, increased range of motion also occurred following massage in the children with burns [52]. Further, burned adolescents have also experienced a reduction in pain and itching following massage [53]. 

Several diseases that have been associated with significant pain that, in turn, has been reduced by massage therapy include juvenile rheumatoid arthritis, sickle cell disease, and chronic pain for no known etiology. This has resulted in pediatric pain clinics offering massage therapy for pain reduction. In a study on sickle cell disease, for example, the children experienced significantly lower levels of pain, anxiety, and depression following massage [54]. In this study, parents gave their children daily 10-min massages. At the end of four weeks, the massaged children had lower levels of pain, anxiety, and depression. In a study on children with juvenile rheumatoid arthritis, parents also massaged their children on a daily basis before bedtime [55]. In this case, pain reduction was reported not only by the parents and children but also by the children’s physicians. 

Several potential mechanisms have been suggested for the relationship between massage and pain reduction. One of the early theories for the effectiveness of massage for alleviating pain is that touch receptors (neurons) are longer and more myelinated (more insulated) than pain receptors and they, therefore, can transmit a signal to the brain faster than pain receptors can. The touch message that is received before the pain message then “closes the gate” (a biochemical/electrical phenomenon), so that the pain message that is the slowest to reach the brain is not received. This is called the “gate theory” [56]. The adult analog is rubbing a bumped “crazy bone”. In a study called “The Gate Theory of Pain Revisited”, a neurocomputational model was formulated that was consistent with the biological one in that pain signals to the brain were blocked when the same area that was “pained” was touched [57]. Stimulation of pressure receptors results in increased vagal activity and serotonin levels (the body’s natural pain suppressor) [58]. A related mechanism is based on the decrease in substance P (a substance known to cause pain) as a result of the decreased cortisol following massage therapy [59]. Furthermore, moderate pressure massage has also been noted to increase oxytocin and decrease beta endorphin [60]. 

### 4.6. Chronic Illnesses

Youth with different chronic illnesses have responded positively to pediatric massage. These include asthma, cystic fibrosis, diabetes, dermatitis, cancer, and HIV. In one of the earliest studies on children with asthma (4–14-year-olds), parents were requested to massage their child for 15 min prior to daily bedtime for four weeks [61]. Based on spirometry on the first and last days of the study, all pulmonary measures improved including forced vital capacity (a 24% increase), forced expiratory volume at one minute (FEV1) (a 27% increase), and peak expiratory flow (a 30% increase) that were recorded every night. These positive effects might have been explained by a significant decrease in cortisol. In a more recent randomized controlled trial on asthma, parents were asked to provide a 20-min massage before bedtime every night for five weeks [62]. At the end of the study, the massaged children as compared to the controls showed higher forced expiratory flow in the first second (FEV1) and a higher FEV1/FVC (forced vital capacity) ratio. Although the groups did not differ on peak expiratory flow, the authors considered FEV1 and FEV1/FVC as the key pulmonary functions, thus concluding that massage therapy significantly benefited these children. In another pediatric massage study on children with asthma, the mothers who massaged their children experienced a decrease in their anxiety levels, highlighting the positive effects for both the massager and the massaged [63].

In a recent systematic review and meta-analysis, 14 randomized controlled studies on 1299 children with asthma were included in the data analysis [64]. In all the studies, peak expiratory flow and FEV1 were significantly increased and decreases were noted in PAF (which produces bronchoconstriction) and prostaglandins (that trigger inflammation). Although the authors highlighted these improvements in pulmonary function, they also noted that the limited research designs led to a high risk for bias. Typically, authors of meta-analyses search the literature for randomized controlled studies that meet inclusion criteria for being selected, e.g., having sufficient power for data analysis. Then the authors assess risk for bias in five domains including selection bias (randomization), performance bias (blinding of treatment), detection bias (blinding of outcome assessment), attrition bias (incomplete outcome data), and reporting bias (selective reporting). If any of these occur, the meta-analysis is considered high risk for bias. Of course, there are difficulties in blinding massage therapy, so those studies are at least at high risk based on that domain.

Cystic fibrosis (a recessive genetic disorder) affects the lungs primarily, although it can also affect the digestive system. Like children with asthma, children with cystic fibrosis have responded positively to massage therapy [65]. In that study on 20 children (5–12-years-old), a reading group was compared to a massage group with the parents either reading to their children or massaging their children for 20 min a night before bedtime for four weeks. Although peak airflow significantly increased, this study did not include primary measures of pulmonary function, and gastrointestinal problems such as abdominal pain were not measured. The authors of a review on several cystic fibrosis studies also reported positive effects of massage [66].

In another chronic illness study, children with diabetes were massaged by their parents for 15 min before bedtime over a period of one month [67]. By the end of the month, the children were showing greater compliance for insulin and food regulation, and their glucose levels had decreased significantly from a high of 159 to 118 within the normal range. In a more recent study, children with diabetes were randomly assigned to a control group or a massage group who received 50-min massages three times a week for three months [68]. Again, the blood glucose levels were significantly lower in the massage group versus a control group by the end of the three-month period. Unfortunately, glycosylated hemoglobin was not measured in either of these studies, a confirmatory measure that might be used in a future study on children with diabetes.

Children with atopic dermatitis (eczema) were massaged as their parents applied medicinal ointment for 15 min a night at bedtime for one month [69]. The control group received the medicinal ointment without massage. The skin condition of the massaged children significantly improved including less redness, lichenification (thickening of skin resulting from scratching), scaling, excoriation (abraded skin), and pruritis (itching).

Several massage therapy studies have been conducted with children with cancer. One of the earliest studies involved parents massaging children who had leukemia 15 min before bedtime on a daily basis for one month [70]. The children who were massaged experienced a decrease in depression and an increase in white blood cells and neutrophils by the end of the study. Surprisingly, none of the studies since then have assessed immune function. In another study on children with cancer, massage was more effective than quiet time at reducing heart rate and anxiety in the children as well as in the parent [71]. In still another study, children with cancer received 20-min massages daily for approximately four days if they were inpatients and once weekly for four weeks if they were outpatients [72]. The massage group as opposed to a standard treatment control group showed a reduction in anxiety, muscle soreness, discomfort and respiratory rate. In a sample of children on a pediatric cancer ward, three 20-min massages were provided on alternate days over a one-week period [73]. The only effects reported were a reduction in pain while walking and during physical activities. Pain and anxiety have also been reduced by massage therapy during bone marrow aspiration in children with cancer [74]. As in most of these studies, pain was assessed on a visual analogue scale and anxiety by a state/trait anxiety scale.

Other measures that have been used to document massage therapy effects include chemotherapy-induced nausea and vomiting and sleep. Because antiemetics are not entirely effective and because of their side effects, massage therapy has been used to reduce chemotherapy-induced nausea and vomiting in pediatric cancer centers [75]. In a randomized controlled clinical trial, 70 children and adolescents undergoing chemotherapy were randomly assigned to a massage group that received 20-min massages 24 h and 0.5 h before chemotherapy and 24 h after chemotherapy [75]. The nausea and vomiting (both the incidence and severity) were significantly less for the massage group. In a sleep study, adolescents with cancer who were hospitalized for at least four consecutive nights were randomly assigned to either a massage or a waitlist control group [76]. The massage group who received at least two massages showed increased nighttime and overall sleep as measured by actigraphy (activity watches), one of the most objective measures used in this literature. 

In a systematic review on massage in children with cancer, the authors found seven articles that met inclusion criteria, although the authors of these studies used different massage techniques and reported on different symptoms [77]. The reviewers concluded that massage decreased pain, nausea, stress, and anxiety. In addition, white blood cells and neutrophils increased, but these were only measured in one of the seven studies, i.e., the study already reported on leukemia [70]. 

## 5. Potential Underlying Mechanisms

The most frequently documented potential underlying mechanism for the massage therapy effects is the increase in parasympathetic activity following stimulation of pressure receptors under the skin [78]. In the earliest demonstration of increased vagal activity or heart rate variability (parasympathetic activity), EKGs were recorded during a three-minute baseline, during a 15-min massage, and three minutes post massage [79]. A group that received moderate-pressure massage was compared to a group that received light-pressure massage. The group that received the moderate-pressure massage showed a shift from sympathetic to parasympathetic activity that peaked during the first half of the massage. The light-pressure massage group, in contrast, showed a sympathetic activity response characterized by decreased vagal activity. This effect was later replicated in a shorter head massage session (10 min) [80]. The heart rate of the group that received massage decreased more than three-fold compared to the control group, and they showed significantly increased vagal activity. The increase in vagal activity was shown in still a third study that compared a medicinal aromatic oil massage group with a sham massage group (light pressure that is known to be arousing) [81]. Heart rate variability (vagal activity) significantly increased in the aromatic oil massage group as compared to the sham massage group. In the only study that explored parasympathetic activity in children, foot and hand massages were given to children who were hospitalized on a pediatric intensive care unit [82]. During the massage, parasympathetic activity (as measured by vagal activity) increased by 75%. 

The increase in vagal activity, in turn, leads to decreased cortisol, and, in turn, increased natural killer cell number and activity. Inasmuch as natural killer cells are noted to kill bacterial, viral, and cancer cells, the enhanced immune function leads to better health. Increased serotonin and decreased substance P also result from enhanced parasympathetic activity. Serotonin, being the body’s natural anti-pain chemical, in turn, knocks down substance P that causes pain. A critical review of 11 studies yielded replicable findings including increased serotonin and natural killer cell activity [83]. The effects on cortisol and immune cells varied according to the amount of pressure, the body site, duration and timing of the massage. Moderate pressure elicited a parasympathetic response in contrast to light touch that elicited a sympathetic response.

## 6. Limitations and Future Directions

Several problems continue to exist for the massage therapy literature. These include the lack of randomized and double-blind trials which make the studies less likely to meet criteria for systematic reviews and meta-analyses. Massage therapy studies are invariably single-blind studies as the participants cannot be blinded to their treatment. Random assignment is not only methodologically difficult but has also become an ethical problem given that massage therapy is known to be effective, and thereby cannot be denied to the research participants in control conditions. Massage therapy effects have often been confounded by the other therapies that accompany the massage, e.g., music and aromas. Similarly, dependent measures have been confounded, for example, correlates of natural killer cells such as pro-inflammatory cytokines need to be measured for their confounding effects on decreased illness. Invasive blood sampling that is still necessary for most immune assays is still difficult for technical and ethical reasons, and blood sampling invariably attenuates the positive effects of massage therapy. However, several pro-inflammatory cytokines can now be assayed in saliva samples. Other measures, for example oxytocin, have not been measured in pediatric massage therapy studies. 

Longitudinal studies are needed to determine whether massage has persistent effects or whether those are reduced or eliminated once the massage is discontinued. In addition, future studies addressing underlying mechanisms are needed to validate the frequent self-report and occasional physiological effects noted for pediatric massage therapy. Despite the methodological limitations of this research, pediatric massage therapy has led to significant improvements in many clinical conditions and has warranted its place as a complementary therapy.

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
