# Peer review of "Pediatric Massage Therapy Research: A Narrative Review"

_children, 2019, doi:10.3390/children6060078_

Round 1
Reviewer 1 Report
Overall this provides a comprehensive overview of massage in children. I particularly like the attention to proposed mechanisms.
However, there is great inconsistency about the level of detail describing the studies mentioned. While i recognize this is comprehensive and we have some limits of space, and that is a narriative review rather than a critical review, it would be very helpful to have more information about the massages and outcomes. Some studies there are details about the duration of massages (ie number of minutes), the frequency and length of the study, and what the comparison group was, and for others there is no such information which makes it hard to know what to make of the conclusions. Similarly for some studies there is information about the outcomes and others it is very generic (for example, was it measured immediately after massage or the next day, or in 3 weeks)
Just one example: "In a similar study on children and adolescents who were hospitalized for their anxiety and depression, those mood disorders were not only decreased but other measures of anxiety and depression including norepinephrine and cortisol were decreased" - how often and for how long were massages delivered? When were the anxiety and depression and lab values measured?
Line by line review:
Intro:please explain how you chose the 48 papers to summarize out of 139 reviewed. There is no explanation about this important aspect of the method.
At the end of the intro it references "table" but i dont see any tables
Section 2: Given how extensive the paper already is, I think it would be better to leave out the pregnancy section, as this is not really related to pediatric massage.
Line 65-66 Other earlier newborn massage studies reported better 66 developmental scores for preterm newborns who had been exposed to cocaine [9] or to HIV in utero - it should say better scores for those who received massage comapred to those who didnt (if this is true)
LIne 110-113 - you state what the review covered but not whether it demonstraed any effects of massage?
Line 145-6 references a table that does not exist..
Line 177 there is a ( that doesnt belong
181-182 (chair massage is preferred by adolescents) - not sure what that is based on
LIne 187 heart rate should be 2 words
Line 206 should say laxative medication use
Line 219-220 "In several studies in an earlier literature and in the more recent literature" - this doesnt make sense
Line 223-224
In a more recent study from Islamabad, Pakistan, 223 children with cerebral palsy (M= six-years-old) - not sure what M=6 yo means ? mean age? is it relevant? you dont list ages for other studies
para starting 268 - i dont believe sickle should be capitalized
Lines 290-92 "A related mechanism is based on substance P decreasing (substance P causing pain) as a result of the decreased cortisol 292 following massage therapy" - this is very confusing - are you trying to say " a realted mechanism is based on the decrease of substance P (a substance known to cause pain)"
Section 5.6 - while there may be some studies linking asthma with autoimmunity, i'm not sure it is generally accepted that asthma is an autoimmune disease. Seems like those 2 paragraphs oculd be separate. Similarly CF is NOT an autoimmune disease so i would not include it in that headline, as affecting automimmuity would not improve CF. As for diabetes, type 1 diabetes is an autoimmune condition, type 2 is not. Were these studies only of type 1 diabetes? in less it said that specifically, i would not include this under autoimmune as the mechanism is unlikely to be related to immune regulation.
LInes 315-19 - i'm not sure you meed to offer a lesson on how meta-analyses are done as part of the paper?
Section 5.7 cancer is multifactorial. since this whole section is on massage in children with cancer, in would label it cancer rather than immune conditions.
End of this section: "Thus, while immune function would seem to be the most critical clinical measure for children with cancer, it hasn’t been included in the existing literature, probably because of limited research funds." I do not agree with this conclusion - there are so many critical clinical measures in children with cancer - i would remove it.
Line 392 -what is a sham massage group?
line 395 parasympathetic activity - as measured by...
LIne 412-14 - Massage therapy is often combined with aroma and music in clinical practice and the research studies in an attempt to be realistic have confounded massage with other therapies
this sentence doesnt make sense..
Author Response
File is attached

Reviewer 2 Report
A brief summary:
This is a nice review of massage therapy in various pediatric uses. The author is clearly an expert in massage therapy based on author’s extensive list of own publications in the field of massage therapy. This author has a good understanding of the field of the massage therapy in pediatrics and in infants and her review is welcomed. There is some room for making this good review an even better one so that the readers of Children will receive a more comprehensive review of massage therapy in pediatric uses.
A few comments I would like to make, and would like the author to address, or correct, when appropriate:
Broad Comments:
1. The author states (in Introduction) “Inasmuch as most of the pediatric massage therapy literature prior to 2008 has not been reviewed (except for the very extensive literature on preterm infant growth following massage), references to earlier studies are given in the text and in tables.”. à Comment: There are total of 27 references that are published prior to year 2008 that the author references. Of these 27 references, 25 are by the author herself and only 2 (references #32, 58) are by someone else than the author. I am aware this author is well-published in the field of massage therapy. Yet it would be important to expand the references to literature prior to 2008 to include articles published by other authors as well.
2. About mechanisms of massage. Comments: The author has a separate dedicated chapter to discuss the possible mechanisms of massage therapy (lines # 377-405). In several chapters author discusses mechanisms already (e.g. lines 135-138 on infant massage, lines 162-167 on attention disorders, lines 213-216 on GI disorders, lines 280-293 on pain, etc.) in the context of the conditions. It would make this review more readable, and also shorter, if mechanisms of massage were either discussed in their respective chapters (with the condition), or all in one chapter where author already discusses some of these mechanisms (at the end of the manuscript lines 377-405), not in both places as is does make readers go back and forth. This change would make the manuscript easier to read and understand. But I do command the author for gathering the information and discussing these mechanisms. Understanding the mechanisms of massage will also make this therapy accepted in all “mainstream” fields of medicine that are typically focused on seeing the hard core “molecular evidence” of any therapy, medicine or intervention. Thank you for including a chapter on mechanisms as well.
Specific Comments:
1. Re full term massage therapy (lines 104-138). A nice comprehensive chapter on massage in full term infants. The author writes: “Since that review was published [8], only a couple papers could be found on full-term newborn massage. One of these was a meta-analysis of randomized controlled trials on reducing bilirubin levels [24].”. Comment: since this is stated to be a review of massage therapies in the past decade, it would make this paper stronger if the author would include the other ones as well (see underlined sentence above) in addition to reference #24.
2. Line # 258. Author writes: “lower state–trait anxiety scores”. Comment: For most readers the expression “state-trait” is likely not clear. This could be briefly explained, defined or referenced in this article as it is a very specific term a reader should understand.
Finally, one general comment I would like to make, given the author being well-known in this field and has a lot of insight into massage therapy: it would be great to have a review (which may be a topic of a separate review article which this author could be an expert for) on different types of massage therapies used in these studies. The fact that many “massage therapy” studies use different forms of massage, with or without oil, and of different duration, may (and likely does) confound results of these studies. To have one review that does not focus on results of the studies but the styles and types of massage therapies, would be welcomed to the audience. This may serve as an invitation or suggestion for this author, who has vast knowledge in this field, to write such a review. Perhaps not the scope of this review though.
If above comments/suggestions are considered (see broad and specific comments), I think this review will add to the field and be welcomed by the readers of Children.
Author Response
File is attached

Reviewer 3 Report
This is a very important paper addressing a therapy that is not widely researched. This format is appealing for those interested in learning more about the evidence.
Author Response
Thank you very much for your compliment on this narrative review.